# Comparative Transcriptome Analysis of *Arabidopsis* Seedlings Under Heat Stress on Whole Plants, Shoots, and Roots Reveals New HS-Regulated Genes, Organ-Specific Responses, and Shoots-Roots Communication

**DOI:** 10.3390/ijms26062478

**Published:** 2025-03-10

**Authors:** Zhaojiao Liu, Xinye Liu, Shuailei Wang, Shuang Liang, Saimei Li, Juntao Wang, Sitong Liu, Yi Guo, Rui Li

**Affiliations:** Ministry of Education Key Laboratory of Molecular and Cellular Biology, Hebei Collaboration Innovation Center for Cell Signaling and Environmental Adaptation, Hebei Research Center of the Basic Discipline of Cell Biology, Hebei Key Laboratory of Molecular and Cellular Biology, College of Life Sciences, Hebei Normal University, Shijiazhuang 050024, China; 13323013388@163.com (Z.L.); xinyeliu@hebtu.edu.cn (X.L.); wangshuailei001@163.com (S.W.); ls2827070654@163.com (S.L.); lisaimei0420@163.com (S.L.); 15612009200@163.com (J.W.); 15226707567@163.com (S.L.)

**Keywords:** aboveground shoots, roots, heat stress, transcriptome analysis, *Arabidopsis*

## Abstract

High temperatures can severely affect plant development and cause a notable decrease in crop yields. Currently, most studies use whole plants that are exposed to steady, high temperatures. This does not reflect the conditions encountered in natural fields, and it overlooks possible differences and coordination between the shoots and roots under heat stress (HS). Here, we analyzed the transcriptome changes in whole plants, shoots, and roots exposed separately to HS. In total, 3346 differentially expressed genes (DEGs) were obtained. Plants in which only the shoots were HS-treated showed minor transcriptional changes compared with whole plants exposed to HS. 62 genes were specifically expressed in HS treatment on shoots, and most of these genes have not been reported to function in HS. We found NAC1 may enhance plant heat tolerance. Utilizing Gene Ontology and Kyoto Encyclopedia of Genes and Genomes analyses, HS-treated shoots showed enhanced gene transcription, protein folding, and MAPK signaling but decreased auxin signaling, while HS-treated roots showed an increase in oxidative stress and suppression of starch and sucrose metabolism. The binding of *cis*-regulatory elements by transcription factors that act downstream in reactive oxygen species (ROS), abscisic acid (ABA), and brassinosteroid (BR) signaling was significantly enriched at the putative promoters of co-expressed genes in shoots and roots under HS treatments on aboveground tissues or roots. Moreover, 194 core HS-responsive genes were identified from all HS treatments, of which 125 have not been reported to function in HS responses. Among them, we found that REV1 and MYC67 may positively regulate the response of plants to heat shock. This work uncovers many new HS-responsive genes and distinct response strategies employed by shoots and roots following HS exposure. Additionally, ROS, ABA, and BR or their downstream signaling factors may be important components for transmitting heat shock signals between shoots and roots.

## 1. Introduction

Temperature is one of the most influential environmental factors known to impact plant growth and development [1,2]. Rising temperatures impact various growth and developmental processes in plants, resulting in such effects as early seed germination, reduced vegetative growth, early flowering, and decreased seed set [1,3]. Heat shock stress perturbs protein folding, membrane fluidity, cytoskeletal organization, cellular transport, and enzymatic reactions, leading to metabolic imbalances and the pernicious accumulation of by-products such as reactive oxygen species (ROS) [4,5]. As the severity of global warming increases, impairments in plant development and crop yields due to heat shock will become more serious [6,7,8]. Therefore, exploring the molecular mechanisms whereby plants respond to and improve their resistance to elevated temperatures will help safeguard global food security.

To cope with heat stress (HS)-associated harm and ensure their fitness, plants have evolved a variety of complex mechanisms to sense and respond to heat shock [9,10]. At the cellular and molecular levels, the main regulatory pathways by which plants respond to high-temperature stress include ROS signaling, phytochrome-B- and phytochrome-interacting factor 4 (PIF4)-mediated pathways, hormone signaling pathways, the Ca^2+^ calmodulin pathway, and an H2A.Z-dependent pathway [11,12]. Our understanding of temperature sensing and signaling has greatly improved over the past decade, mainly from studies in which whole plants have been subjected to HS with no separate consideration of the aboveground shoots and roots. However, shoots and roots differ in their morphology and physiological functions. Roots usually grow at much lower temperatures than aboveground shoots, and they apply distinct physiological and metabolic changes to adapt to high temperatures [13,14,15].

Still, a few studies have focused on the different responses of the aboveground/underground organs of plants to elevated temperatures. For example, PIF4 has been identified as a key component in hypocotyl and petiole elongation at warm temperatures; however, normal heat-induced root elongation has been observed in *pif4* mutant plants, implying that the mechanisms underlying thermomorphogenesis in roots and aboveground shoots are distinct [16,17,18]. Detached aboveground shoots and roots have been used to explore the responses of these plant parts to HS [14,19,20]. The shoots, hypocotyls, and roots of *Arabidopsis thaliana* (hereafter, *Arabidopsis*) seedlings were found to have specific transcriptome responses to warm temperatures, and detached roots could sense and respond to temperature independent of shoots [14]. Indeed, roots are sensitive to heat stress, which seriously inhibits cell division in the root meristem and impairs root growth [15]. However, when roots were cultivated in a device that generated a temperature gradient mimicking that found in natural fields, healthy shoot growth was observed, and plant biomass production increased at elevated ambient temperatures [15]. Meanwhile, the levels of several plant hormones, including salicylic acid (SA), ethylene (ET), and abscisic acid (ABA), increase under HS, conferring tolerance to hot temperatures. These hormones are candidate signaling molecules that may mediate communication between roots and aboveground tissues [21]. Thus, given the different responses of shoots and roots to high temperatures and the complexity of shoot-root signaling, traditional homogeneous HS treatment of whole plants cannot reflect the actual growth conditions of heat-shocked aboveground shoots and roots. Furthermore, a large number of key genes, including heat shock factor/protein (HSF/HSP) genes [22], play critical roles in the response of plants to HS; however, whether these proteins are specifically expressed in aboveground organs, roots, or both is unknown. Exploring their expression separately in aboveground shoots and roots will help decipher their functional mechanisms under heat shock stress.

A transcriptome analysis of plants under HS is an effective strategy to discover crucial HS-responsive genes. However, most transcriptome studies have been performed using plants exposed to homogeneous high temperatures. To identify new key factors that respond to early HS in plants growing in natural fields, to explore the different responses of aboveground shoots and roots during early HS, and to investigate the transduction of HS signals between shoots and roots, three different HS treatments were applied to *Arabidopsis* seedlings: whole-plant HS at 42 °C (42W), HS applied to either the aboveground shoots or roots of whole plants at 42 °C (42S or 42R, respectively), and whole seedlings exposed to 22 °C as a control (Figure 1A). Subsequently, a comparative transcriptomic analysis was performed for whole-plant exposed to HS, aboveground shoots, and roots exposed separately to HS. We found HS exposure of the aboveground shoots resulted in fewer DEGs than did HS treatment of the whole plant. Overall, 62 genes were found to be specifically expressed following such treatment. Different GO and KEGG pathways were enriched among the DEGs from the aboveground shoots and roots. ROS, ABA, and BR or their downstream factors may be important signaling molecules for transducing heat shock signals. Additionally, 194 core genes were identified from among the co-expressed genes identified from all three types of HS treatment.

## 2. Results

### 2.1. Transcriptomic Data Were Obtained from the Aboveground Shoots and Roots of Arabidopsis Seedlings Following Heat Shock Treatment of the Whole Plant, Shoots, or Roots

To selectively heat shock the aboveground shoots and roots of plants, a double-deck temperature control device (TCD) was created (Figure 1B). The roots of 15-day-old *Arabidopsis* seedlings were immersed in the lower layer of the TCD’s water-circulating box at a temperature of either 42 °C or 22 °C (Figure 1A,B). The water-circulating box was placed in a temperature-regulated chamber set at 42 °C or 22 °C to control the temperature of the aboveground shoots (Figure 1A,B). Prior to our large-scale transcriptomics assay, the expression of the HS marker gene *HSFA2* was probed in plants subjected to the aforementioned HS treatments using RT-qPCR (Figure 1A,C). *HSFA2* showed high expression both in shoots and roots of plants under the 42W treatment (Figure 1C). In the 42S and 42R groups, *HSFA2* expression was also significantly increased in the tissues exposed to HS; however, the levels were considerably lower than those observed for the 42W samples (Figure 1C). *HSFA2* was also induced in non-stressed roots and shoots following 42S or 42R treatment (Figure 1C). Therefore, the expression levels of *HSFA2* in shoots and roots were distinct among the three HS treatments.

Next, a transcriptome analysis was conducted using *Arabidopsis* (ecotype Columbia [Col]-0) seedlings cultivated on half-strength Murashige and Skoog (1/2MS) plates for 15 days. The four groups of seedlings were, respectively, exposed to three different HS treatments with whole plants grown at 22 °C (22W) as a control (Figure 1B). Subsequently, the aboveground shoots and roots were frozen in liquid nitrogen for RNA extraction and RNA sequencing (RNA-Seq) with three biological replicates (Figure 1A). In total, 1.04 billion high-quality reads were generated using the Illumina sequencing platform, which were then mapped to the wild-type Col-0 reference genome using HISAT2 [23]. On average, ~96% of the reads were uniquely mapped (Appendix A), and only these uniquely mapped reads were used to calculate normalized gene expression levels, expressed as Transcripts Per Kilobase of exon model per Million mapped reads (TPM). The expression level for each experimental sample was determined based on the average TPM value from the three biological replicates. To mitigate the impact of transcriptional noise, we defined a gene as expressed if its TPM value was ≥1.

To gain insight into the transcriptome dynamic of aboveground shoots and roots across four different treatment groups, we performed hierarchical clustering analysis and principal component analysis (PCA) for the eight samples. The eight datasets clustered into two distinct groups corresponding to the transcriptomes of aboveground shoots and roots (Figure 1D,E). This supports the reported differences in physiology and development between shoots and roots under HS [14]. The transcriptome samples of aboveground shoots were further subdivided into two branches; samples from the HS treatment of whole plants (42W-S) and selective HS treatment of shoots (42S-S) clustered together (Figure 1D,E). Samples from the control group (22W-S) and roots treated selectively with HS (42R-S) were grouped (Figure 1D,E), suggesting that 42R treatment had less impact on shoots. Similarly, the transcriptome samples for roots were divided into two branches, the control group (22W-R) and the three HS-treated groups, indicating that HS treatment notably affected gene transcription in the roots (Figure 1D,E). Furthermore, the 42W-R and 42S-R samples were grouped together, while 42R-R had its own group, indicating unique transcriptome dynamics. The expression levels of the HS marker genes *HSFA2* and *HSFA7A*, as determined by RT-qPCR, were consistent with the results of our hierarchical clustering analysis (Appendix A), confirming the reliability of our transcriptome data.

To systematically investigate HS-responsive transcriptional dynamics in shoots and roots across three HS experiments, we identified organ-specific differentially expressed genes (DEGs) through comparative transcriptome profiling between three HS-treated groups and the control group (Figure 1F). Using a *q*-value of ≤0.05 and a fold change ≥2 or ≤0.5 to screen for differentially expressed genes (DEGs), we found 3346 DEGs in the HS-exposed samples (Appendix A), including 266 transcription factor genes (Appendix A). The numbers of up- and down-regulated DEGs were quantified for each HS treatment (Figure 1F). Treatment 42R-R produced the highest number of DEGs (2108), consistent with its distinct transcriptomic profile. Among the shoot samples, 42W-S had the highest number of DEGs (951), higher than the 520 DEGs for 42S-S. Abundant DEGs were detected in both 42S-R and 42R-S samples, suggesting that HS signals can be transmitted to the untreated organs of plants following HS treatment of shoots or roots alone.

### 2.2. The Transcriptomic Profiles of the Aboveground Shoots Were Significantly Different from Those of the Roots in Whole Plants Subjected to Early HS Treatment

Before undertaking a comprehensive analysis of the DEGs in our 42W-S and 42W-R samples, the expression levels of four HS marker genes (*HSFA2*, *HSFA7B*, *HSP18.2*, and *HSP18.5*) were assessed using whole plants subjected to HS. Consistent with our RNA-Seq data, these genes were strongly induced in both the 42W-S and 42W-R samples, with notably higher expression levels in the shoots, further confirming the reliability of our transcriptomic data (Appendix A). Subsequently, we compared the DEGs from the shoots and roots of whole plants subjected to HS. A total of 951 DEGs were identified in the shoots (520 up- and 431 down-regulated), which significantly exceeded the 440 DEGs found in the roots (247 up- and 193 down-regulated) (Figure 1F and Figure 2A). Only 191 DEGs (151 up- and 33 down-regulated) were shared between the shoots and roots (Figure 2A), indicating that distinct mechanisms are employed in these tissues to adapt to elevated temperatures. Furthermore, among the 191 shared DEGs, 7 had opposite expression trends in the shoots and roots (Table 1). Such differences could easily be missed in traditional transcriptomic analyses using homogeneously heated whole plants (i.e., no significant change would be detected). For instance, *BAM9*, which encodes a starch hydrolase, and the ABA-induced gene AT5G23350 were up-regulated in the shoots but down-regulated in the roots, while the cytokinin response factor gene *ARR6* and UDP-glucosyltransferase gene were down-regulated in the shoots but up-regulated in the roots (Table 1). These findings support the notion that shoots and roots may activate or inhibit specific physiological processes in opposing manners in response to HS.

*HSFB1*, *NAC029*, *ERF053*, and *WRKY30* play critical roles in the HS response in *Arabidopsis*; however, it remains unclear whether these genes exhibit distinct expression patterns in shoots or roots under HS conditions [8,29,30,31]. In our transcriptomic data, *HSFB1* was induced in both shoots and roots, with a higher expression level observed in shoots (Figure 2B), suggesting its potential contribution to the HS response of the whole plant. Conversely, *NAC029* and *ERF053* were exclusively up-regulated in shoots, while *WRKY30* was specially down-regulated in roots (Figure 2B), indicating their unique roles in shoots or roots.

Subsequently, we conducted Gene Ontology (GO) enrichment analysis on all upregulated and downregulated differentially expressed genes (DEGs) in the shoots and roots of plants subjected to the 42W treatment. The functional categories were ranked based on their enrichment significance, and the top ten categories with a *q*-value ≤ 0.05 were selected for visualization (Figure 2C,E and Appendix A). Among the up-regulated DEGs, the shoots and roots exhibited significant enrichment in “regulation of transcription, DNA-templated” and “protein folding” (Figure 2C). Notably, the functional categories “unfolded protein binding” and “response to stress” were exclusively enriched in the aboveground shoots. This indicates that shoots induce the expression of a greater number of stress-responsive genes. Furthermore, the roots showed significant enrichment in the functional categories “NADH dehydrogenase activity”, “ATP metabolic process”, and “generation of precursor metabolites and energy” (Figure 2C). This suggests that roots utilize antioxidant mechanisms and respiratory processes to alleviate the damage caused by elevated temperatures. Significant differences were observed between shoots and roots in the downregulated DEGs (Figure 2D). “Protein-DNA complex” and “response to hormone” were enriched in the shoots, whereas “cell wall” and “enzyme inhibitor activity” were enriched in the roots.

Subsequently, a Kyoto Encyclopedia of Genes and Genomes (KEGG) metabolic pathway enrichment analysis was conducted (Figure 2E,F and Appendix A). “Protein processing in the endoplasmic reticulum” was the most prevalent pathway in the shoots and roots; this encompasses numerous molecular chaperones that facilitate the proper folding of proteins under HS. Notably, for the up-regulated DEGs, the “spliceosome” pathway and “oxidative phosphorylation” pathway, respectively, were enriched in shoots and roots (Figure 2E and Appendix A). This observation further suggests that roots increase ATP synthesis to sustain cellular homeostasis in response to high-temperature stress.

Among the down-regulated DEGs, and similar to our GO analysis, the metabolic pathway “plant hormone signal transduction” was significantly enriched in shoots, but not in roots (Figure 2F and Appendix A). In shoots, many genes involved in the auxin signaling pathway were down-regulated, while ABA-responsive genes were up-regulated. However, only three negative regulators of the cytokinin signaling pathway were up-regulated in roots (Appendix A). Thus, hot temperatures inhibit the auxin signaling pathway, especially in shoots. Additionally, the pathways “phenylpropanoid biosynthesis” and “starch and sucrose metabolism” were significantly enriched in roots (Figure 2F), further suggesting a reduction in metabolic activity in roots under HS.

In summary, during whole-plant HS treatment, a significant number of transcription factors, heat shock proteins, and genes associated with protein folding were activated in both shoots and roots, with more genes being activated in the shoots. Most of the activated pathways in the roots were related to ATP metabolism and antioxidant responses. Auxin signaling pathways were primarily inhibited in the shoots, while genes related to the cell wall and the synthesis of secondary metabolites were inhibited mainly in the roots.

### 2.3. 62 DEGs Were Specifically Expressed Following HS Treatment of Shoots

In the 42S treatment groups, 520 DEGs were identified in the shoots (386 up- and 134 down-regulated), and 94 DEGs were found in the roots (89 up- and 5 down-regulated) (Figure 3A). Notably, 80 up-regulated DEGs were shared between the shoots and roots, accounting for most (80/89) of the up-regulated genes in the roots; these could be induced by the same upstream signaling pathway. Interestingly, no gene was co-down-regulated in the shoots and roots.

A comparative analysis of the DEGs between the 42S and 42W treatment groups was conducted (Figure 3B,C). The number of DEGs in the 42S-S samples was significantly lower than that in the 42W-S samples, particularly for down-regulated genes. In the shoots, 486 genes (150 up- and 336 down-regulated DEGs) were specifically expressed in the shoots of heat-shocked whole plants. In the roots, 86 genes were up-regulated in both the 42S and 42W samples, whereas only 1 gene was co-down-regulated (Figure 3C). This difference may come from the effects of HS treatment on roots and/or the additive effects of HS treatment on whole plants.

Additionally, 62 genes (16 up- and 39 down-regulated DEGs in 42S-S; 3 up- and 4 down-regulated DEGs in 42S-R) were specifically expressed in the 42S samples but were not found following whole-plant HS treatment (Figure 3B,C). Regarding the shoots, eight of the up-regulated genes have been documented to play a role in the response to other abiotic stresses, while another eight up-regulated genes have not been linked to any stress response (Table 2). Further, 39 DEGs were identified as down-regulated in the shoots, of which 11 have unknown functions. The remaining DEGs were primarily associated with abiotic stress responses and plant hormone signal transduction pathways (Appendix A). Notably, WRKY26 has been documented to positively influence heat tolerance in *Arabidopsis* [32]. In roots, two genes, *AtFKBP62* and *TPR10*, have been implicated in heat stress via interactions with HSP70/90 (Table 2). Four genes (*NAC1*, *DEG11*, *PLAC8*, and *AT1G55430*) were specifically down-regulated in the roots (Table 3). Among these, the transcription factor NAC1 is implicated in auxin-mediated lateral root formation (Table 3). To investigate the function of NAC1 under HS, we obtained the knock-down T-DNA mutant *nac1* (SALK_052190c) from the AraShare stock center https://www.arashare.cn/ (accessed on 1 March 2025) (Appendix A). Following growth at 22 °C for 4 days, the germinated seeds were exposed to either 22 °C (homogeneous whole-plant treatment at 22 °C) or 28 °C (homogeneous whole-plant treatment at 28 °C) for 6 days (Figure 3D). Compared to wild type, the number of emerged lateral roots (eLR) and the biomass of the *nac1* mutants were remarkably increased (Figure 3E,F), but the length of the primary root was unaffected (Appendix A). Thus, NAC1 may play a role in plant thermomorphogenesis.

GO functional enrichment analysis showed significant enrichment exclusively among the up-regulated genes of the 42S-S group (Figure 3G and Appendix A), while no significant enrichment was detected in the down-regulated genes from the 42S-S or 42S-R samples. Similar to the up-regulated genes from the 42W-S group, there was marked enrichment in the functional categories of “DNA binding transcription factor activity” and “protein folding”. The 42S-S samples showed significant enrichment in the category of “ATPase regulator activity” and “chaperone binding”, which were not found in 42W-S. Conversely, the functional categories “regulation of transcription, DNA-templated”, “aromatic compound synthesis”, and “response to stress”, which were enriched in 42W-S, were absent from the 42S-S group (Figure 2B,C and Figure 3G). These findings suggest that HS treatment of shoots stimulates ATP production and chaperone binding activity, whereas HS treatment of whole plants triggers more defensive responses and antioxidant processing.

KEGG enrichment analysis revealed that, similar to 42W, “protein processing in endoplasmic reticulum” and “MAPK signaling pathway” were enriched pathways among the up-regulated genes of the shoots and roots following 42S treatment (Figure 2E and Figure 3H, and Appendix A). The up-regulated DEGs from the 42S shoot and root samples showed specific significant enrichment in “endocytosis” (Figure 2D,E and Figure 3H). For the down-regulated DEGs, similar to the 42W-S samples, “plant hormone signal transduction” and “glucosinolate biosynthesis” were the most enriched pathways in the shoots of 42Streated plants (Figure 2F and Figure 3I). The down-regulated DEGs from the 42S-S samples showed significant enrichment in the “linoleic acid metabolism” pathway, which was not found in the 42W-S samples (Figure 2F and Figure 3I) [31]. Due to the limited number of DEGs in the 42S-R samples, no significant enrichment of any pathways was noted.

In summary, the application of heat shock to aboveground shoots elicits a relatively mild response characterized by a reduced number of DEGs and increased ATP hydrolase, chaperone activity, and endocytosis, which are absent from the DEGs of HS-treated whole plants.

### 2.4. The Transcriptional Response in Roots of Plants Exposed to Root-Specific HS Was Much More Pronounced than That in Roots from Whole Plants Exposed to HS

Previous research has indicated that detached seedling roots are capable of sensing and responding to elevated temperatures [14]. However, the mechanisms by which the roots of intact plants respond to HS signals are poorly understood. Root-specific HS treatment of whole plants was undertaken in our paper. Although such a scenario is hardly encountered in natural fields, it has significant importance for elucidating the responses of roots to HS signals and understanding the effects of root HS on aboveground shoots.

In total, 2108 DEGs (589 up- and 1519 down-regulated) were identified in our 42R-R samples, which is many more than were detected in our 42W-R samples (Figure 4A). Overall, 358 DEGs were shared between the 42R-R and 42W-R groups, but these represented only 17% (358/2108) of the total number of DEGs in the 42R-R samples. However, they accounted for 81.2% (358/440) of the total number of DEGs in the 42W-R group (Figure 4B). Thus, the heat response of the roots in the 42R group was more pronounced than that in the 42W group.

By GO enrichment analysis, the up-regulated DEGs in the 42R-R samples showed significant enrichment in the categories of “generation of precursor metabolites and energy”, “electron transport chain”, “proton transmembrane transport”, and “protein folding”, which is different from those in the 42W-R samples; still, “NADH dehydrogenase activity” and “ATP metabolic process” were enriched in the 42R-R samples, as in the 42W-R samples (Figure 2B,C and Figure 4C, and Appendix A). The down-regulated DEGs in the 42R-R samples exhibited notable enrichment in “phosphotransferase activity, alcohol group as acceptor”, “oxidoreductase activity, acting on a sulfur group of donors”, and “copper ion binding” (Figure 4D), unlike the 42W-R samples. Furthermore, “cell wall” and “aspartic-type endopeptidase activity” were enriched in both the 42W-R and 42R-R samples (Figure 2B,C and Figure 4D).

By KEGG analysis, the up-regulated DEGs in the 42R-R samples were significantly enriched in “oxidative phosphorylation”, “protein processing in endoplasmic reticulum”, and “MAPK signaling pathway”, like those in the 42W-R samples (Figure 4F and Appendix A). Surprisingly, some pathways that were specifically enriched in the aboveground tissues were among the DEGs from the 42R-R samples. Spliceosome-related genes were enriched among the up-regulated DEGs from the 42W-S and 42R-R samples, while alpha-linolenic acid metabolism-related genes were enriched among the down-regulated DEGs from the 42S-S and 42R-R samples (Figure 4G). Similar to 42W-R, the down-regulated DEGs from 42R-R were highly enriched in “phenylpropanoid biosynthesis” and “starch and sucrose metabolism”, while many more pathways, including “ABC transporters”, “plant hormone signal transduction”, “MAPK signaling pathway-plant”, and “galactose metabolism”, were also represented among the DEGs from the 42R-R samples.

Therefore, the transcriptional changes observed in the 42R-R samples were considerably more dynamic than those observed in the 42W-R samples, and 42R treatment induced alterations in a broader range of biological processes within roots.

A total of 353 DEGs (286 up- and 67 down-regulated) were identified in the 42R-S samples (Figure 4A). Only the up-regulated DEGs from the 42R-S samples showed significant GO enrichment, and the enrichment patterns were different from those of the 42S-S and 42W-S samples (e.g., “histone acetyltransferase activity” and “zinc ion binding”). KEGG enrichment analysis revealed that the up-regulated genes from the 42R-S samples were enriched in “starch and sucrose metabolism”, which was repressed in the roots of both the 42W-R and 42R-R groups. These findings suggest that heat-stressed roots transmit heat signals to the aboveground shoots and activate distinct processes and pathways. endocytosis, which are absent from the DEGs of HS-treated whole plants.

### 2.5. The Co-Induced DEGs in Shoots and Roots Following Shoot- or Root-Specific HS Treatment May Be Regulated by Transcription Factors Acting Downstream of ROS, ABA, and BR Signaling

Although it is believed that HS signals can be transduced between aboveground shoots and roots [14,15,46], the signaling molecules that mediate this process are less understood. Following HS treatment of just the aboveground shoots or roots in whole plants, the untreated organs exhibited dynamic transcriptional changes, confirming that heat signals can be transduced to untreated tissues. In total, 129 DEGs were co-induced in shoots and roots in the 42S and 42R treatment groups (Figure 3A and Figure 4A). Among these, 54 core genes were induced in both the 42S and 42R treatment groups, while 26 genes were specifically induced in the 42S samples and 49 genes were specifically induced in the 42R samples (Figure 5A). To find the signaling molecules responsible for this intercommunication between shoots and roots under HS, we analyzed the *cis*-elements in the putative promoters of these co-induced DEGs. About 1000 bp of genomic sequence located upstream of the translation start codons (ATG) for the co-induced DEGs were subjected to Multiple Em for Motif Elicitation (MEME) analysis http://meme-suite.org/tools/centrimo (accessed on 1 March 2025). Nearly half (47.3%) of the genes were found to be potentially regulated by HSFA4A, followed by HSFA1E, ABA responsive element-binding factor 2 (ABF2), ABA insensitive 5 (ABI5), HSFB2B, basic region/leucine zipper transcription factor 68, HSFA6A, ABA-induced serine-rich transcription repressor (ASR1), and G-box binding factor 2 (GBF2) (Figure 5B).

The HSFA4A binding motif was the most highly enriched motif among the 54 core genes and 49 genes specific to the 42R samples (Figure 5C,E). It has been reported that the accumulation of ROS under HS activates the mitogen-activated protein kinase 4/6/3 (MPK4/6/3) pathway, which then phosphorylates and activates HSFA4A, regulating the expression of its downstream target genes [47]. Therefore, the rapid accumulation of ROS induced by high temperatures may be an important event mediating HS signaling in both shoots and roots.

In the ABA signaling pathway, ABA activates serine/threonine protein kinases (SnRK2s), which then phosphorylate ABF2/ABI5 transcription factors to activate downstream ABA-responsive genes and regulate leaf senescence [48]. ABF2, ABI5, HSFA6A, and ASR1 are key transcription factors in the ABA signaling pathway, and their binding motif appeared frequently in 129 genes (Figure 5C–E). Among them, *cis*-elements bound by ABF2 and ABI5 were specifically detected with high abundance in the DEGs from the 42S and 42R samples (Figure 5D,E), while *cis*-elements recognized by HSFA6A and ASR1 were exclusively found in 54 core DEGs. In plants, the root cap and leaves are the main sites of ABA synthesis, and following its synthesis, ABA is distributed rapidly to neighboring tissues [49]. Thus, ABA may be an important candidate HS signaling molecule responsible for transduction between shoots and roots. In addition, regulatory elements recognized by the BR-responsive transcription factors GBF2 and BEE2 were present in these co-induced DEGs (Figure 5D,E). Therefore, ROS, ABA, and BR may serve as signaling molecules for HS in shoots and roots to enhance the overall heat tolerance of plants.

### 2.6. A Total of 194 Core DEGs Responsive to Early HS Were Identified Using Three HS Treatments

To identify the core genes that respond to early heat shock, a Venn diagram was created using the DEGs identified in aboveground shoots and roots exposed to our three HS treatments. In total, 194 core genes were expressed under all HS conditions. Among them, 109 DEGs and 32 DEGs were specifically expressed in aboveground shoots and roots, respectively, while 53 DEGs appeared in both (Figure 6A). GO enrichment analysis of these core DEGs showed significant enrichment in the functional categories “response to heat”, “cellular response to hypoxia”, and “protein folding” (Figure 6B). Overall, 69 of the 194 DEGs have been reported to be implicated in the response to HS (including 41 HSFs/HSPs) (Appendix A), and 59 of the 194 DEGs are involved in responses to other abiotic stresses such as cold, salt, and drought. Surprisingly, 66 of the 194 DEG genes have not been linked to any abiotic stress (Appendix A). Of the 53 core DEGs expressed in both aboveground shoots and roots, 15 have not been reported to be involved in HS responses (Appendix A). These core genes should be investigated further.

We selected the Myb-like transcription factor REVEILLE1 (RVE1), which integrates the circadian clock and auxin pathways [50], and a MYC-type transcription factor 67 (MYC67), which negatively regulates cold-responsive genes in cold tolerance [51], and studied their potential roles in HS responses. RT-qPCR confirmed that the expression of *RVE1* was significantly up-regulated in both shoots and roots, while *MYC67* was down-regulated only in shoots following each of the three HS treatments (Figure 6C). We next obtained homozygous null mutants of *rve1* and *myc67* from the AraShare stock center https://www.arashare.cn/ (accessed on 1 March 2025). (Appendix A). Seven-day-old seedlings grown at 22 °C were allowed to recover for 7 days after HS treatment (45 °C for 60 min) (Figure 6D,F). Compared to wild type, the survival rates of the *rve1* and *myc67* seedlings were significantly reduced, indicating a notable decrease in heat tolerance (Figure 6E,G).

## 3. Discussion

Aboveground organs and roots in natural ecosystems face different temperatures under HS. The shoots of plants are directly subjected to high temperatures, whereas the roots are deeply embedded in the soil, which offers short-term protection from direct exposure to elevated temperatures [15,52]. A recent study demonstrated that a low temperature in the root ecosystem sustained plant growth under HS. When the aboveground shoots of *Arabidopsis* seedlings were exposed to a high temperature while the roots were exposed to gradual high-temperature stress, the plants were able to grow effectively and maintain their functionality to support optimal shoot growth and development, whereas whole plants exposed to a single high temperature showed impaired root growth and a lower ability to adapt to HS [15]. Thus, heat shocking the underground parts of plants can seriously damage plant growth. However, to date, most transcriptional analyses have relied on whole-plant HS treatment to identify genes involved in high-temperature responses. Our study explored transcriptional dynamics, organ-specific responses, and complex shoots-roots communications and discovered core HS-regulated genes through HS application to whole plants or to shoots/roots alone. We identified multiple new genes involved in heat shock, and we uncovered differences and coordination between aboveground organs and roots at elevated ambient temperatures.

### 3.1. Comparative Analysis of the Transcriptome Between HS-Treated Whole Plants and HS-Treated Aboveground Shoots or Roots

In our study, treatment 42W induced more pronounced dynamic changes in gene expression in aboveground shoots compared to treatment 42S, with significant increases in the numbers of up- and down-regulated DEGs. Notably, the number of down-regulated genes for treatment 42W-S was three times the number for treatment 42S-S (Figure 1F and Appendix A). GO and KEGG analyses revealed that treatment with 42S specifically enhanced ATP metabolism and endocytosis in HS-treated aboveground shoots. Conversely, treatment 42W resulted in the up-regulation of a greater number of genes associated with transcriptional regulation, antioxidant activity, and stress responses. Furthermore, aboveground shoots subjected to both treatments exhibited a response to elevated temperatures that included the inhibition of signaling pathways associated with plant hormones, including auxin and ABA; however, the number of down-regulated genes was lower than under treatment 42S. Treatment 42S also specifically inhibited linoleic acid metabolism (Figure 2 and Figure 3). These findings suggest that HS treatment of whole plants may lead to an exaggerated response to high temperatures that ultimately inhibits cell growth. This is consistent with the conclusions drawn from the transcriptome data and plant growth status of seedlings observed under long-term whole-plant HS treatment and gradient high-temperature treatment of roots [15]. These data indicate that the application of both short- and long-term HS to whole plants has a significantly greater impact on plant physiological processes and gene expression than heat shock applied only to aboveground shoots. This further implies that maintaining lower temperatures around the roots may enhance a plant’s ability to cope with high temperatures, resulting in less detrimental effects on plant growth and development. This phenomenon could be explained by the fact that HS signals are transduced from the roots to the aboveground shoots in HS-treated whole plants, thereby eliciting a more robust stress response in the shoots, or it may result from the additive effects of simultaneously heat shocking the shoots and roots.

Whole-plant HS treatment exhibited significantly more upregulated and downregulated genes compared to HS application to shoots alone. These DEGs may not be induced or suppressed in plants growing in natural fields under heat stress conditions. Notably, 62 genes exhibited specific responses to HS in the 42S treatment group (Appendix A). *NAC1* encodes a transcription factor involved in auxin-mediated lateral root development [44]. Our preliminary experiments indicate that the number of lateral roots in *nac1* increased significantly at a warm temperature, suggesting its potential role in the regulation of plant thermomorphogenesis. Other genes expressed in the 42S-S samples should be examined further to explore their function in HS. For instance, *Chaperone DnaJ-domain type C 53* (*DjC53*) is up-regulated in aboveground shoots and has been identified as a negative regulator of plant thermotolerance [35]. Additionally, it has been reported to participate in light responses, functioning downstream of ELONGATED HYPOCOTYL 5 (HY5) [34]. Genes encoding twin cysteine proteins play a critical role in the response of plants to abiotic stresses, including drought and light [36].

We also conducted a comparative analysis of the transcriptomic data from the 42W and 42R treatment groups. Notably, a greater number of DEGs were found in the 42R-R samples, with a significant increase in down-regulated genes (Figure 4). This observation is consistent with the finding that stress-exposed roots could activate defensive responses in all organs, with the most substantial gene expression changes occurring in the roots [13]. This phenomenon may be partially explained by the fact that the exposure of roots to HS rarely happens in natural fields; thus, plants have not evolved adaptive strategies to cope with this condition. Alternatively, it is possible that whole-plant HS inhibits gene expression or reduces the sensitivity of roots to HS (Figure 1F). Consequently, whole-plant HS treatment should not be regarded merely as the addition of HS to the roots and aboveground shoots; rather, it may suppress gene expression in the roots while simultaneously enhancing gene expression in the shoots and facilitating a more effective response to high temperatures. To clarify the effects of HS exposure in roots on plant growth and development, plants in which the roots have been selectively exposed to HS should be monitored for growth and compared with plants in which the entire plant has been exposed to HS.

### 3.2. Different Response Strategies to Heat Shock Are Employed in Aboveground Shoots and Roots

Previous work has found differences in the responses of aboveground shoots and roots to elevated temperatures during long-term heat treatment of whole plants [14,15]. The DEGs in shoots subjected to prolonged heat stress at 32 °C were predominantly enriched in such categories as photosynthesis, light signaling, auxin signaling, heat response, and protein folding. In contrast, the DEGs in roots exposed to HS showed substantial enrichment in categories related to HS, water deprivation, RNA modification, oxidative stress, and hypoxic responses [15]. The results of our PCA and clustering analysis indicate that the transcriptomic data from aboveground shoots and roots were categorized into two distinct branches, suggesting significantly different characteristics of shoots and roots under HS (Figure 1). Furthermore, following whole-plant HS treatment, the differences between shoots and roots were manifested as variations in the number of DEGs. Notably, the number of DEGs in shoots was substantially greater than that in roots, with only a limited number of DEGs being commonly expressed in both tissues (Figure 2). Treatment regimen 42S may mimic the situation of plants in natural fields, and the number of DEGs in shoots was significantly higher than that in roots (Figure 3). Among the 194 core DEGs identified for all HS treatments, the number of DEGs in shoots was markedly greater than that in roots. These data suggest that shoots exhibit more dynamic gene expression changes in response to HS compared to roots, positioning shoots as the primary organs for coping with heat shock. Finally, based on GO and KEGG analyses of the DEGs across the three HS treatments, it was observed that both shoots and roots can activate protein processing in the endoplasmic reticulum. However, shoots and roots employ distinct mechanisms in response to HS. Shoots predominantly respond to HS by enhancing gene transcription, protein folding, and the MAPK signaling pathway, while inhibiting the auxin signaling pathway. In contrast, roots primarily induce oxidative stress and suppress starch and sucrose metabolism, cell wall formation, and enzyme activity.

### 3.3. The Reciprocal Influence and Intercommunication Between Aboveground Shoots and Roots in Response to Heat Shock

In our study, treatments 42S and 42R, respectively, led to significant changes in the number of DEGs in untreated roots and shoots. When subjected to 42S treatment, only a few DEGs were found in the roots. Conversely, 42R treatment had a more profound impact on the gene dynamics of shoots (Figure 1F). Conversely, in our cluster analysis of eight samples following the four treatments, the 42S-R and 42W-R samples clustered together, while the 42R-S and 22W-S samples formed a separate cluster. This suggests that HS treatment of shoots has a more pronounced effect on gene transcription in roots (Figure 1E). Clustering analysis may yield more reliable results since heat signals in natural environments are predominantly transmitted from aboveground shoots to roots. However, our understanding of the molecules that transmit heat signals between aboveground shoots and roots is poor. It has been proposed that the plant hormones SA, ABA, and ET function as significant signaling molecules, facilitating communication between shoots and roots [53,54]. Local heating of cucumber shoots or roots resulted in an increase in ABA concentration not only within the heated organ but also in non-heated tissues, suggesting that ABA is an important signal between shoots and roots [55]. Furthermore, previous studies have found that auxin can serve as a long-distance transport signal, moving from leaves to the hypocotyl to promote hypocotyl elongation [14]. Our results indicate that following treatments 42S and 42R, nearly half of the commonly induced genes in shoots and roots contained regulatory elements that are bound by HSFA4A, which acts downstream of ROS signaling in response to HS [47]. However, whether ROS serves as critical molecules for heat signal transmission between shoots and roots is unknown. Future experiments could inhibit the production of ROS in shoots to determine whether HSFA4A-related genes are affected in untreated tissues. Regarding treatments 42S and 42R, numerous genes possessed regulatory elements for the transcription factors ABF2, ABI5, and GBF2, which function downstream of ABA and BR signaling [48,56]. Unlike HSFA4A, the genes containing elements bound by ABF2, ABI5, and GBF2 were different between the 42S and 42R groups, indicating that ABA and BR or their downstream factors may migrate to non-heat-shocked areas to induce the expression of different genes. Measurement of the ABA and BR levels in non-heat-shocked regions is necessary to support this hypothesis.

In summary, our study utilizes a comparative transcriptomics strategy to investigate the differential responses of shoots and roots to whole-plant HS versus HS treatments on shoots or roots individually. Our work demonstrates that HS treatments on shoots alone induce fewer DEGs compared to whole-plant HS and also identify new heat-responsive candidate genes (*NAC1*, *MYC67*, and *RVE1*). The analysis further reveals distinct adaptive strategies employed by aerial and underground tissues under HS, alongside potential signaling molecules mediating shoots-roots communication, including ROS, ABA, and BR. These findings provide critical insights into plant HS adaptation mechanisms and highlight the sophisticated and well-coordinated communication between shoots and roots to cope with HS.

## 4. Methods

### 4.1. Plant Materials

In this work, we used *Arabidopsis* ecotype Col-0. Mutants with the following code numbers were obtained from the AraShare stock center: SALK_052190C (NAC1 AT1G56010), SALK_009770 (MYC67 AT3G61950), and SALK_025754 (RVE1 AT5G17300).

### 4.2. Stress Treatment and Sample Collection

*Arabidopsis* seedlings were germinated on MS1/2 medium containing vitamins plus 1% sucrose and 0.6% Gelzan™ CM (pH 5.7–5.8) and grown under long-day conditions (16 h of light/8 h of dark). A TCD was used to generate different temperatures in the aboveground shoots and roots of plants. The water circulation box of the TCD had upper and lower layers, featuring a rectangular hole in between to fit the 12 × 12 cm external plate. The roots of the seedlings, which were positioned on the culture plate, were situated in the lower layer of the water circulation system; their temperature was maintained by a constant temperature water circulation mechanism. Concurrently, the aboveground shoots of the seedlings were situated in the upper gaseous environment. The entire water circulation box was housed within a temperature-regulated chamber, allowing for precise control of the temperature experienced by the aboveground shoots. The aboveground shoots and roots were subjected to two temperature conditions, 22 °C (control) and 42 °C (heat shock), using the TCD system. Seedlings were germinated in MS1/2 for 15 days at 22 °C and then transferred to 22 °C (22W), 42 °C (42W), 42 °C in the upper layer, and 22 °C in the lower layer with separation by an iron cover (42S), or 22 °C in the upper layer and 42 °C in the lower layer with separation by an iron cover (42R) for 30 min.

The stress-treated and control samples were compared at the same time point: the conclusion of the heat shock period. For each sample, roots or shoots from a minimum of 15 plants were collected, frozen in liquid nitrogen, and stored at −80 °C. This experiment was replicated three times to ensure reliable results.

### 4.3. RNA Isolation and Assessment

A total of 24 samples [2 tissues (shoots and roots) × 4 treatments (control and TCD device) × 3 biological replicates] were used in our transcriptome analysis. Total RNA was extracted using an RNeasy Plus Mini Kit (Agilent Technologies, Santa Clara, CA, USA); RNA contamination and degradation were assessed on 1% agarose gels. RNA integrity was assessed using an RNA Nano 6000 Assay Kit and an Agilent 2100 Bioanalyzer (Agilent Technologies).

### 4.4. Illumina Transcriptome Library Preparation, Sequencing, and Data Analysis

The RNA was quantified using a Qubit 2.0 RNA High Sensitivity Assay Kit (Q32855; Thermo Fisher Scientific, Waltham, MA, USA). A transcriptome mRNA library was constructed using a Hieff NGS™ MaxUp Dual-mode mRNA Library Prep Kit for Illumina^®^ (12301ES96; Yeasaen, Shanghai, China). The amplified products of the library were purified using Hieff NGS™ DNA Selection Beads (12601ES56; Yeasen) (0.9×, Beads:DNA = 1:1). A Qubit 2.0 DNA Detection Kit (Q10212; Thermo Fisher Scientific, Waltham, MA, USA) was used to quantify the recovered DNA to facilitate sequencing after equal mixing at a ratio of 1:1. For Illumina sequencing, an indexed library of 24 internodal RNA samples was prepared and sequenced using the Illumina HiSeq™ platform (Illumina Inc., San Diego, CA, USA).

The original image data file from the Illumina HiSeq™ system was converted into sequenced reads by base calling using CASAVA from Illumina Inc.; the results were stored as FASTQ files. The raw data in FASTQ format were controlled according to the scripts of Sangon Biotech Co., Ltd. (Shanghai, China). In this step, clean data were obtained by removing three kinds of reads: those containing adaptors, low-quality reads, and reads with fewer than 35 nucleotide bases. Meanwhile, the Q20 and GC content were used to estimate the quality of these clean data.

### 4.5. Quantification of Gene Expression Levels and DEG Analysis

The transcript expression levels were identified using StringTie 1.3.3b software and the TPM method for each sample [57]. DEGs were identified using a *q*-value < 0.05 and log_2_(Group1/Group2) ≥ 1 as the threshold values in pairwise comparisons of the experimental and control samples. The DEGs were used to identify potential transcription factors from the PlantTFDB database http://planttfdb.cbi.pku.edu.cn/ (accessed on 1 March 2025).

### 4.6. Functional Enrichment Among the DEGs

GO https://www.geneontology.org/ (accessed on 1 March 2025) and KEGG http://www.kegg.jp (accessed on 1 March 2025) enrichment was determined using the clusterProfiler package in R.3.3.0 The *Arabidopsis* Genome Initiative codes for genes with a |log_2_FC| > 1 and *q*/*p*-value < 0.05 were analyzed for different gene sets [58]. In general, when the *q*- or *p*-value is <0.05, it is concluded that there is significant enrichment of this function (Appendix A).

### 4.7. Differential Gene Cis-Element Enrichment

Enrichment of *cis*-elements in the −1000 to −1 promoter sequences of the 129 DEGs (downloaded from TAIR10; http://www.arabidopsis.org/, accessed on 1 March 2025) was performed using the online tool MEME (version 5.5.7; https://meme-suite.org/meme/tools/ame, accessed on 1 March 2025) [59].

### 4.8. RT-qPCR Analysis

RT-qPCR was conducted using a CFX96 Touch™ Real-Time PCR Detection System (Bio-Rad, Hercules, CA, USA) with SYBR^®^ Green and analyzed with CFX Manager 3.1 software (Bio-Rad). The PCR program was as follows: 95 °C for 10 min and 33 cycles of 95 °C for 15 s and 60 °C for 30 s. Specific primers were designed using Primer3 4.1.0 as shown in Appendix A [60]; their specificity was confirmed by a BLAST (https://blast.ncbi.nlm.nih.gov/, accessed on 1 March 2025) search of common *Arabidopsis* transcripts. As an internal standard, the *Arabidopsis* ubiquitin gene was selected to calculate relative fold expression levels according to the threshold cycle method.

## Figures and Tables

**Figure 1 ijms-26-02478-f001:**
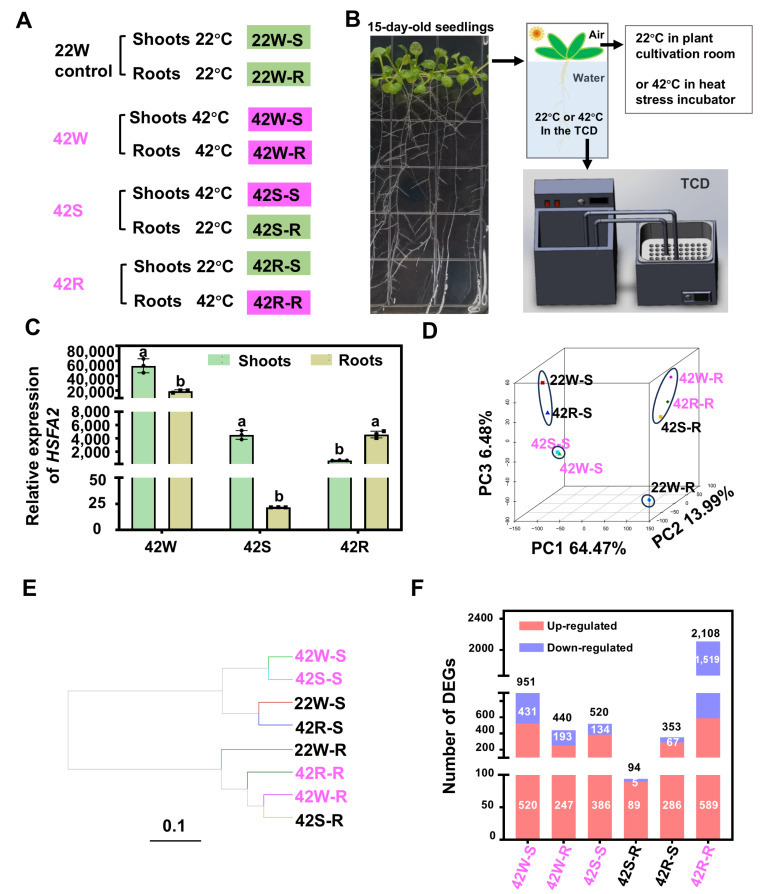
Notable transcriptomic differences among the heat shock-treated plants depending on whether heat was applied to the whole plant, aboveground shoots, or roots. (**A**) The aboveground shoots and/or roots of *Arabidopsis* seedlings were subjected to HS as follows. 22W-S/R: After growth at 22 °C (control), either the aboveground shoots or roots were collected as samples. 42W-S/R: After whole-plant HS treatment (42 °C), the aboveground shoots/roots were collected as samples. 42S-S/R: After HS treatment (42 °C) of just the aboveground shoots, the aboveground shoots/roots were collected as samples. 42R-S/R: After HS treatment (42 °C) of just the roots, the aboveground shoots/roots were collected as samples. W: whole plant; S: aboveground shoots; R: roots. (**B**) Schematic diagram of the TCD used for the heat shock treatment of the aboveground shoots and/or roots of the plants. HS was administered to the aboveground shoots and roots utilizing a light culture chamber and a circulating water tank maintained at 42 °C, respectively. (**C**) *HSFA2* expression in 15-day-old seedlings treated in the TCD was observed by RT-qPCR. *UBC21* was used as the internal control. Each dot represents the result from one biological replicate; error bars indicate the mean ± standard error (SE). Statistically significant differences are indicated by different lowercase letters (*p* < 0.05, two-way ANOVA with Tukey’s significant difference test). (**D**) PCA of the transcriptomes of the eight experimental samples. (**E**) A cluster dendrogram showing two distinct patterns: aboveground shoots and roots. (**F**) The differentially expressed genes were obtained by comparing the transcriptional data from experimental groups of shoots and roots with those of the control group. Statistics on the number of DEGs from different pairwise comparisons. *q*-value ≤ 0.05 and a fold change ≥2 or ≤0.5.

**Figure 2 ijms-26-02478-f002:**
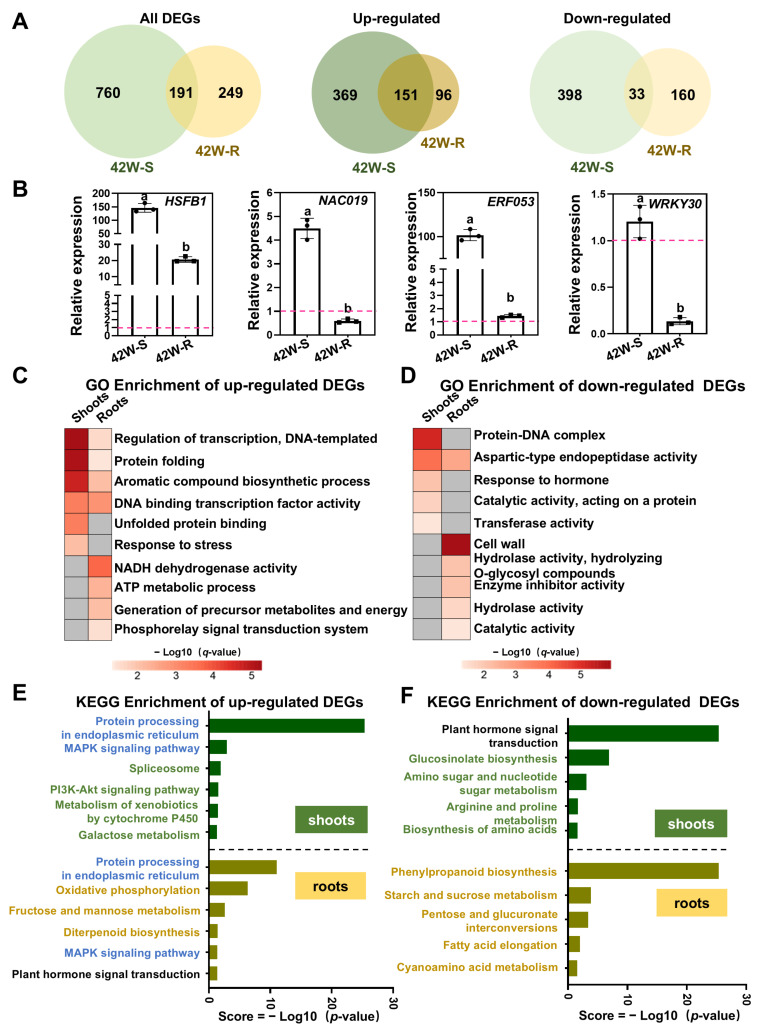
The DEGs in the aboveground shoots and roots of whole plants subjected to HS function in different metabolic pathways as a response to elevated temperatures. (**A**) Venn diagrams of the DEGs for different comparisons (up- or down-regulated genes) based on the 42W-S or 42W-R samples. (**B**) The expression levels of specifically responsive transcription factors in shoots or roots after whole-plant HS. The pink dotted line indicates the control group gene expression level. *UBC21* was used as the internal control. Each dot represents the result from one biological replicate; error bars indicate the mean ± SE. Statistically significant differences are indicated by different lowercase letters (*p* < 0.05, two-way ANOVA with Tukey’s significant difference test). (**C**,**D**) Selection of the terms biological processes (BP), cell component (CC), and molecular function (MF) based on a GO enrichment analysis of the up- (**C**) and down-regulated (**D**) DEGs after whole-plant HS. The color of the column represents the range of the *q*-value; gray represents NA (*q*-value < 0.05). (**E**,**F**) Histogram representation of the KEGG pathways for the up-/down-regulated DEGs in the 42W samples. The figure displays only significantly enriched metabolic pathways (*p*-value ≤ 0.05), sorted in descending order of enrichment. Green/Yellow text indicates the functional categories that are specifically enriched in up/down-regulated DEGs of 42W-S/R; Blue text indicates the functional categories that are both enriched in up-regulated DEGs at 42W-S and 42W-R; the black text indicates the signaling pathway, which is enriched among upregulated genes in the roots, is also enriched among downregulated genes in the shoots.

**Figure 3 ijms-26-02478-f003:**
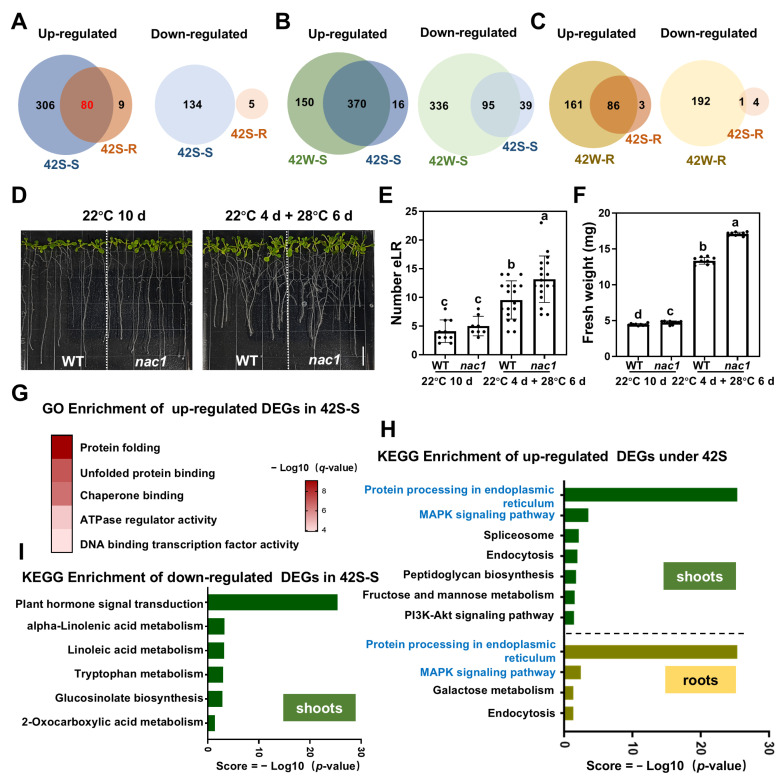
Selective HS treatment of aboveground shoots resulted in specific gene expression in response to high-temperature stress compared with whole-plant HS treatment. (**A**) Venn diagrams showing the DEGs for different comparisons (up- or down-regulated genes) of the 42S-S or 42S-R samples. (**B**,**C**) Venn diagrams showing the DEGs for different comparisons (up- or down-regulated genes) of the 42W or 42S samples. (**D**) Representative images of 10-day (d)-old *Arabidopsis* seedlings grown at 22 °C (22W) and 28 °C (4 d 22 °C + 6 d 28 °C). Bar = 1 cm. (**E**) Quantification of the number of eLR. (**F**) Quantification of the fresh weight per seedling. Each dot represents the result from one biological replicate; error bars indicate the mean ± SE. Statistically significant differences are indicated by different lowercase letters (*p* < 0.05, two-way ANOVA with Tukey’s significant difference test). (**G**) GO enrichment analysis of the DEGs from the 42S samples. Red text indicates functional categories that were specifically enriched among the DEGs of the 42S-S samples (*q*-value < 0.05). (**H**,**I**) Histograms showing the results of our KEGG enrichment pathway analysis of the up- (**H**) and down-regulated (**I**) DEGs from the 42S samples. The figure displays only significantly enriched metabolic pathways (*p*-value ≤ 0.05), sorted in descending order of enrichment. Blue text indicates the functional categories that are both enriched in up-regulated DEGs at 42S-S and 42S-R (*p*-value < 0.05).

**Figure 4 ijms-26-02478-f004:**
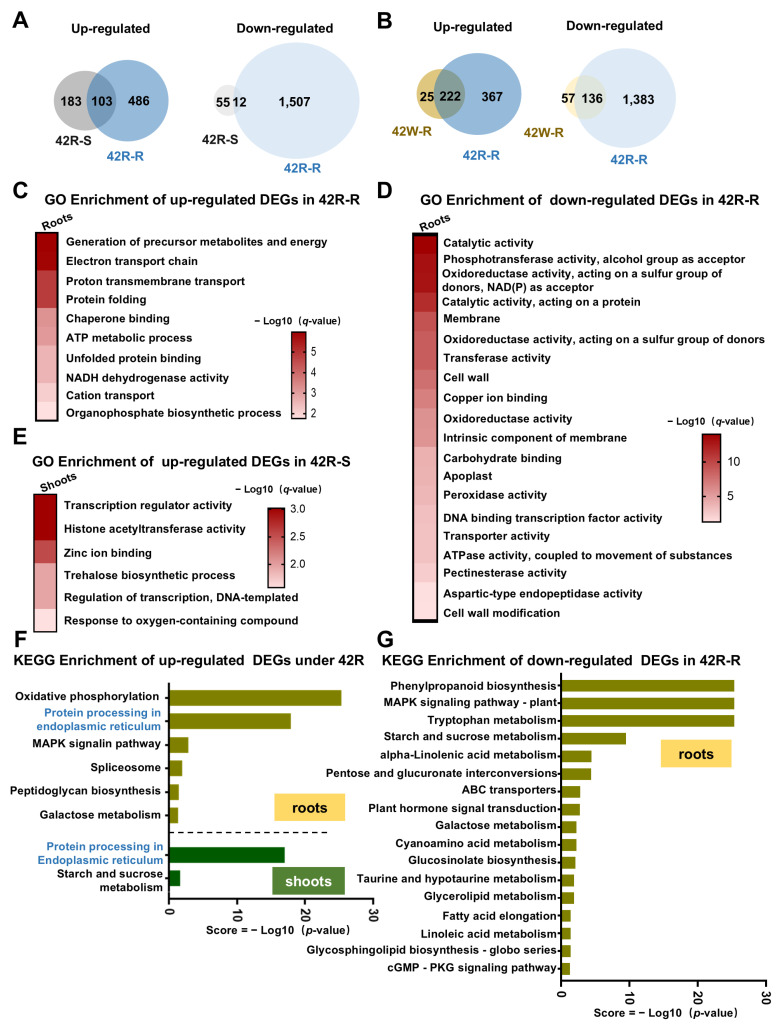
A greater number of HS-responsive genes were identified in plants where only the roots were heat-stressed compared with whole-plant HS treatment. (**A**) Venn diagrams showing the DEGs for different comparisons (up- or down-regulated genes) of the 42R-S or 42R-R samples. (**B**) Venn diagrams showing the DEGs for different comparisons (up- or down-regulated genes) of the 42W and 42R groups. (**C**,**D**) GO term enrichment among the up- and down-regulated DEGs from the 42R-R samples (*q*-value < 0.05). (**E**) GO term enrichment among the up-regulated DEGs in the 42R-S samples. Functional classes written in bold showed the same level of enrichment in the 42W-S samples (*q*-value < 0.05). (**F**,**G**) Histograms showing KEGG pathway enrichment for the up- and down-regulated DEGs from the 42R samples. The figure displays only significantly enriched metabolic pathways (*p*-value ≤ 0.05), sorted in descending order of enrichment. Blue text indicates the functional categories that are common for up-regulated DEGs at 42R-R and 42R-S.

**Figure 5 ijms-26-02478-f005:**
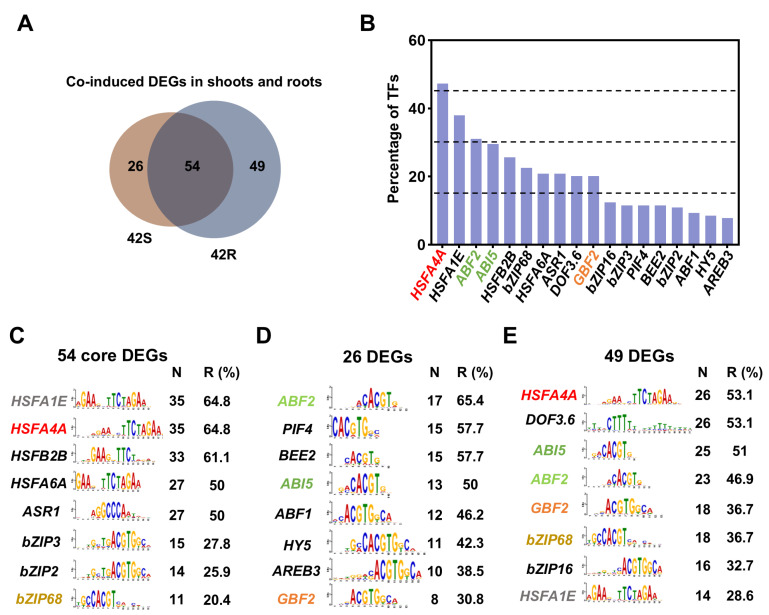
Identification of transcription factors involved in high-temperature signal transmission between aboveground shoots and roots. (**A**) DEGs that were co-induced in both shoots and roots under 42S or 42R treatment. (**B**) Enrichment of *cis*-elements recognized by various transcription factors in the 1000-bp promoter region of HS-up-regulated genes in both shoots and roots. (**C**–**E**) The top eight most highly enriched *cis*-elements of the genes in the three groups in (**A**). N: Number; R: Ratio.

**Figure 6 ijms-26-02478-f006:**
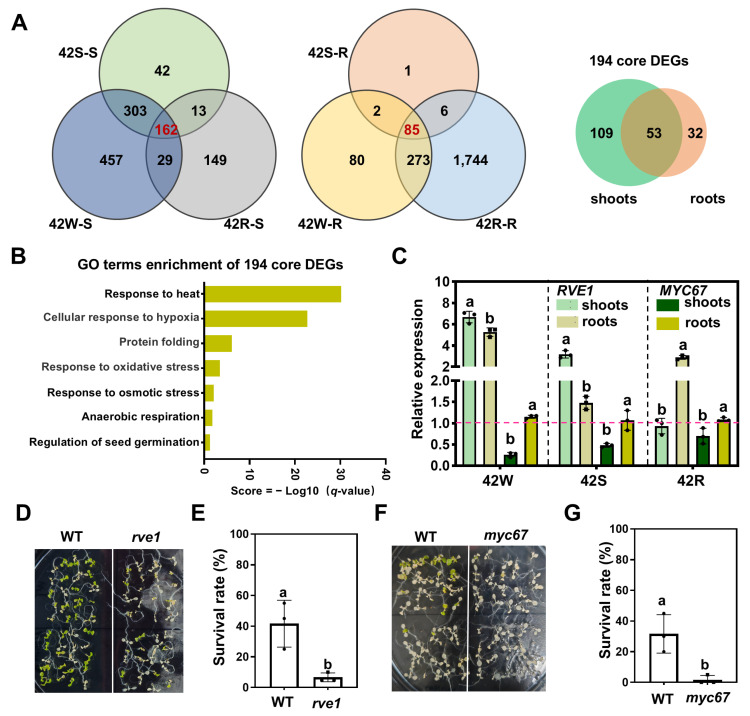
Core genes involved in high temperature regulation across the three experimental groups. (**A**) Venn diagrams of the DEGs in different comparisons of the shoot or root samples. (**B**) Histogram showing the GO terms assigned to the shoot and root core genes, *q*-value ≤ 0.05. (**C**) *RVE1* and *MYC67* expression under different HS treatments. *UBC21* was used as the internal control for RT-qPCR. The pink dotted line indicates the control group gene expression level Each dot represents the result from one biological replicate; error bars indicate the mean ± SE. Statistically significant differences are indicated by different lowercase letters (*p* < 0.05, two-way ANOVA with Tukey’s significant difference test). (**D**–**G**) Comparison of thermotolerance using wild-type (WT), *rve1*, and *myc67* plants. Seven-day-old *Arabidopsis* seedlings grown at 22 °C under continuous light were used. Seedlings that were allowed to recover for 7 days after HS at 45 °C for 60 min were photographed, and the survival rates of the plants were calculated. Each dot represents the result from one biological replicate; error bars indicate the mean ± SE. Statistically significant differences are indicated by different lowercase letters (*p* < 0.05, two-way ANOVA with Tukey’s significant difference test).

**Table 1 ijms-26-02478-t001:** DEGs that showed opposite trends in the 42W-S and 42W-R samples.

Locus Tag	Regulation	Regulation	Official Full Name	Description
	(42W-S)/FC	(42W-R)/FC		
AT5G18670	Up/3	Down/−3.7	BAM9	b-Amylase BAM9 regulates starch breakdown, and its gene expression is responsive to several environmental changes [24].
AT4G32480	Up/6	Down/−2.5	-	Phosphorus (P) stress-inducible DUF506 gene family member [25].
AT5G23350	Up/4.3	Down/−2.3	ABA-responsive-like protein	GRAM domain protein/ABA-responsive-like protein
AT3G16560	Down/−2.4	Up/2.25	Protein phosphatase 2C family protein	This gene belongs to PP2C subfamily C and is induced by cold and heat [26].
AT5G01712	Down/−4	Up/12	-	Unknown
AT5G62920	Down/−3	Up/4.4	Type-A response regulator 6	A Type-A response regulator that is responsive to cytokinin treatment and negative regulator of cytokinin [27].
AT3G21560	Down/−3.1	Up/2	UDP-GLUCOSYL TRANSFERASE 84A2	*brt1* showed smaller stomatal apertures than wild type under normal light conditions and UV-B irradiation [28].

**Table 2 ijms-26-02478-t002:** Specifically up-regulated DEGs in the 42S compared with the 42W samples.

Locus Tag	Official Full Name	Localization	Description
42S-S			
Stress response			
AT5G16110	Hypothetical protein	Nucleus	Related to abiotic stress [33].
AT1G13600	bZIP-58	Nucleus	Downstream target gene of HY5 [34].
AT1G56300	Chaperone DnaJ-domain protein	Nucleus	DJC53 may play a negative regulatory role in response to HS [35].
AT5G64400	Twin cysteine proteins	Mitochondrion	The loss of both *At12cys-1* and *At12cys-2* leads to enhanced tolerance to drought and light stress and increased antioxidant capacity [36].
AT1G50740	Transmembrane proteins 14C	Extracellular	Cellular response to hypoxia [37].
AT5G42220	Ubiquitin-like protein	Nucleus	Misfolded protein binding and polyubiquitin modification-dependent protein binding [38].
AT1G29260	PEX7	Peroxisome	Under certain stressful conditions, such as in the presence of high amounts of H_2_O_2_ produced within peroxisomes, PEX7 may be damaged and thus accumulate on the peroxisomal membrane [39].
AT3G62740	Beta glucosidase 7	Extracellular	The expression of AtBGLU7 was significantly up-regulated when treated with cold for 24 h [40].
Other and unknown			
AT4G30200	VEL1 protein	Nucleus	VEL1 is a part of a polycomb repressive complex (PRC2) that is involved in epigenetic silencing of the FLC flowering locus [41].
AT1G78150	N-lysine methyltransferase	Chloroplast	Unknown function
AT4G19430	Hypothetical protein	Mitochondrion	Unknown function
AT3G57810	Cysteine proteinases protein	Mitochondrion	Unknown function
AT2G43630	Nucleus envelope protein	Plastid	Unknown function
AT1G30190	Cotton fiber protein	Nucleus	Unknown function
AT1G14970	O-fucosyltransferase protein	Nucleus	Unknown function
AT1G21680	DPP6 N-terminal domain-like protein	Extracellular	Unknown function
42S-R			
Stress response			
AT3G25230	FK506 binding protein 62	Cytosol	Modulates thermotolerance by interacting with HSP90.1 and affecting the accumulation of HSFA2-regulated HSPs [42].
AT3G04710	Tetratricopeptide repeat 10	Cytosol	Encodes 1 of the 36 carboxylate clamp (CC)-tetratricopeptide repeat (TPR) proteins with the potential to interact with HSP90/HSP70 as co-chaperones [43].
AT3G13470	Chloroplasts chaperonins	Plastid	Encodes a subunit of chloroplast chaperonins (CHAPERONIN-60BETA2, CPNB2).

**Table 3 ijms-26-02478-t003:** DEGs that were specifically down-regulated in the 42S-R compared with the 42W-R samples.

Locus Tag	Official Full Name	Localization	Description
Plant hormone signal transduction			
AT1G56010	Transcription factor NAC1	Nucleus	Encodes a transcription factor involved in auxin-mediated lateral root formation, auxin-activated signaling pathways, and lateral root development [44].
Light signaling pathway			
AT2G34430	Photosystem II type I chlorophyll a/b-binding protein	Plastid	Light harvesting in photosystem II [45].
Unknown			
AT1G49030	PLAC8 family protein	Plasma membrane	Unknown function
AT1G55430	Cysteine/histidine-rich C1 domain protein	Nucleus	Unknown function

## Data Availability

RNA-seq data that underpin the findings of this study have been deposited in the Sequence Read Archive (SRA) of the National Center for Biotechnology Information (NCBI) under the bioproject accession number PRJNA1205352 (http://www.ncbi.nlm.nih.gov/bioproject/1205352, accessed on 1 March 2025). By searching the provided accession code, one can access the uploaded RNA-seq data.

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
