# Peer review of "Comparative Transcriptome Analysis of Arabidopsis Seedlings Under Heat Stress on Whole Plants, Shoots, and Roots Reveals New HS-Regulated Genes, Organ-Specific Responses, and Shoots-Roots Communication"

_ijms, 2025, doi:10.3390/ijms26062478_

Round 1
Reviewer 1 Report
Comments and Suggestions for Authors
The manuscript is well written and contains a detailed description of the Arabidopsis transcriptomic analysis experiment under heat shock. I think that it guides the reader well through the stages of the experiment: from the design of the device for controlled heat stress, through the analysis of the expression of marker genes, to the RNA-Seq analysis and clustering results.
Statistical and bioinformatic methods were used to evaluate gene expression (q-value ≤ 0.05, fold change ≥ 2), which strengthens the reliability of the results. I think that the manuscript highlights well the differences in gene expression between plant organs (roots and shoots) and the fact that the heat shock signal can be transferred to untreated parts of the plant.
Despite the above advantages, I think that the paper requires some additions and corrections. I have attached some suggestions below:
1. Some passages are very long and complex and could be written in a more accessible way:
- ‘HSFA2 was highly expressed in both the aboveground shoots and roots of plants that received treatment 42W (Fig. 1C).
- ‘Our study applied HS to whole plants and to shoots or roots alone to explore the different transcriptional dynamics between homogeneous and selective HS, how aboveground organs and roots respond differently to HS, the intercommunication between shoots and roots, and to identify core heat shock genes.’
2. The manuscript should be analysed and changes made to improve the flow between chapters and sections, e.g. as in 2.1. - the transition from PCA analysis to differentially expressed genes (DEGs) and the sudden jump to numerical results.
3. Please clarify, or explain better, whether the statement ‘whole-plant HS treatment may generate a considerable amount of transcriptional noise’, suggesting that this type of HS can falsify the results, refers to greater variability in gene expression or to the difficulty in interpreting the plant response?
4. The paper does not have a general summary.
Reviewer 2 Report
Comments and Suggestions for Authors
The manuscript is well-structured and logically organized, with high readability. Addressing following points will enhance its rigor and impact.
- In Fig. 1F, the differential genes are presented; however, the article and figure legend do not specify how the differential groups were established. This information is crucial for understanding the screening of differential genes and should be included in the figure legend, methods section, or main text.
- For the GO enrichment analysis graphs presented in Fig. 2 (C and D), what data were utilized to create them? The main text states that the top 10 up-regulated and down-regulated differentially expressed genes (DEGs) were used. Is this right?
- Regarding the KEGG enrichment analysis bar charts in Fig. 2 (E and F), are all the pathways depicted significantly enriched based on the p-value? In addition to the green, yellow, and blue colors in the figure legend that indicate specific and common DEG-enriched pathways, what does the black color represent?
- In the Venn diagram shown in Fig. 6 (A), it is advisable to ensure that all circles are of equal size to enhance clarity and visual balance.
5. The title of this manuscript is too long.
